# Gut Bacteriomes and Ecological Niche Divergence: An Example of Two Cryptic Gastropod Species

**DOI:** 10.3390/biology12121521

**Published:** 2023-12-13

**Authors:** Elizaveta Gafarova, Dmitrii Kuracji, Karina Sogomonyan, Ivan Gorokhov, Dmitrii Polev, Ekaterina Zubova, Elena Golikova, Andrey Granovitch, Arina Maltseva

**Affiliations:** 1Department of Invertebrate Zoology, St. Petersburg State University, 199034 St. Petersburg, Russia; dkuracij@gmail.com (D.K.); zubova.15@list.ru (E.Z.); a.granovich@spbu.ru (A.G.);; 2Center for Bioinformatics and Algorithmic Biotechnology, St. Petersburg State University, 199034 St. Petersburg, Russia; karinasog96@gmail.com; 3Department of Epidemiology, St. Petersburg Pasteur Institute, Mira Street 14, 197101 St. Petersburg, Russia; brantoza@gmail.com

**Keywords:** bacterial microbiomes, 16S, shotgun, *Littorina*, cryptic species, symbiosis

## Abstract

**Simple Summary:**

Nowadays, it is well known that the gut bacterial microbiome is crucially important for the adaptation of multicellular organisms to their environment. In this study, we aimed to identify the differences between the bacteriomes of two closely related marine snail species living in sympatry but feeding in different micro-niches. Although there were significant interspecies differences detected during the summer season, we did not observe this dissimilarity during the cold season. Moreover, the diversity of bacterial communities associated with snails decreased in autumn. We suggest that bacteria predominantly associated with one species degrade its toxic feeding substrate. These results help to understand the role of bacteriomes in the adaptation and divergence of closely related species.

**Abstract:**

Symbiotic microorganisms may provide their hosts with abilities critical to their occupation of microhabitats. Gut (intestinal) bacterial communities aid animals to digest substrates that are either innutritious or toxic, as well as support their development and physiology. The role of microbial communities associated with sibling species in the hosts’ adaptation remains largely unexplored. In this study, we examined the composition and plasticity of the bacteriomes in two sibling intertidal gastropod species, *Littorina fabalis* and *L. obtusata*, which are sympatric but differ in microhabitats. We applied 16S rRNA gene metabarcoding and shotgun sequencing to describe associated microbial communities and their spatial and temporal variation. A significant drop in the intestinal bacteriome diversity was revealed during the cold season, which may reflect temperature-related metabolic shifts and changes in snail behavior. Importantly, there were significant interspecies differences in the gut bacteriome composition in summer but not in autumn. The genera *Vibrio*, *Aliivibrio*, *Moritella* and *Planktotalea* were found to be predominantly associated with *L. fabalis*, while *Granulosicoccus*, *Octadecabacter*, *Colwellia*, *Pseudomonas*, *Pseudoalteromonas* and *Maribacter* were found to be mostly associated with *L. obtusata*. Based on these preferential associations, we analyzed the metabolic pathways’ enrichment. We hypothesized that the *L. obtusata* gut bacteriome contributes to decomposing algae and detoxifying polyphenols produced by fucoids. Thus, differences in the sets of associated bacteria may equip their closely phylogenetically related hosts with a unique ability to occupy specific micro-niches.

## 1. Introduction

Microorganisms associated with metazoans can powerfully influence the adaptive scope of their hosts. The gut (intestinal) microorganisms of both vertebrates and invertebrates can supply the host with enzymes, antimicrobials, nutrients and other metabolites [1,2,3]. The survival of a multicellular organism or its feeding specialization may directly depend on cooperation with bacteria [1,2,4,5,6]. There is evidence that microorganisms participate in the digestion of non-nutritive and even toxic substrates. For instance, the snails *Rubyspira osteovora* live in the deep-sea whale falls and feed on whale bones. The intestinal microbial community provides these snails with the ability to ferment the bone tissue [6]. The pest species of coffee plantations, the beetle *Hypothenemus hampei*, feeds on coffee bushes, despite the presence of the toxic alkaloid caffeine. Ceja-Navarro et al. [4] showed that caffeine can be degraded in the gut of *H. hamppei* and that experimental inactivation of the gut microbiota eliminates this ability. Bacterial symbionts may also be involved in the proper development of the digestive, immune, circulatory and nervous systems [7,8,9,10,11]. Moreover, prokaryotic symbionts significantly change the physiology and behavior of their hosts [12,13,14,15,16,17,18]. This explicitly demonstrates the critical importance of analyzing associated bacteriomes when studying how metazoans adapt to their ecological niches. Intestinal bacteria are an important mediator of hosts’ adaptation to the conditions of the inhabited niche, which has been shown in vertebrate [19] and invertebrate [20,21] animals.

Regarding recently diverged sympatric sister-species, their ecological specialization to different parts of the environment contributes to their reproductive isolation. In addition, it provides an advantage in alleviating interspecies competition and promoting more efficient exploitation of environmental resources [22,23,24]. The subdivision of ecological niches within the same biocoenosis requires different adaptive complexes in young species, which may be dependent on the associated bacteriome [20,21,25,26,27,28,29]. Nevertheless, the evolutionary fate of the associated microbial community during ecological speciation is still poorly investigated. Here, we analyzed the composition and plasticity of the bacterial microbiomes in two sister-species of intertidal gastropods (Caenogastropoda, Mollusca). 

Marine intertidal mollusks from the genus *Littorina* are used as an informative model for studies on invertebrate evolutionary biology, ecology and adaptation [30,31,32]. Atlantic snails of the subgenus *Littorina* (*Neritrema*) have been comprehensively studied in the context of post hoc and ad hoc sympatric speciation events along environmental gradients [33]. While various trends accompanying divergence at the levels of whole genomes [34] and single genes [32] of the *Littorina* snails have been broadly investigated, their bacteriomes had been nearly ignored until recently. Over the past few years, general descriptions of bacterial microbiome analysis of the *Littorina* (*Neritrema*) species [35] and the ecotypes of *L. saxatilis* [36] were published. Moreover, a hypothesis was suggested about the role of symbionts in the reproductive barriers’ formation in this group of species [37]. Our study focuses on two sister-species within *L.* (*Neritrema*): *L. fabalis* and *L. obtusata.*

*Littorina fabalis* (Turton 1825) and *L. obtusata* (Linnaeus 1758) (Littorinidae: *Littorina*: *Neritrema*) are intertidal micro-grazers living together on the rocky shores of the North Atlantic [38]. Even being similar morphologically and physiologically, these species show different patterns in micro-biotopic distribution. *L. fabalis* prefers to stay on *Fucus serratus* in the lower intertidal and the upper subtidal zones while *L. obtusata* prefers *Fucus vesiculosis* and *Ascophyllum nodosum* at the lower and the middle intertidal levels. Although the distributions of these species partially overlap [38,39,40,41,42], they are considered to be informative models for understanding the factors, mechanisms and consequences of ecological speciation [35,42,43].

In this study, we applied 16S rRNA gene metabarcoding and shotgun sequencing to compare the bacteriomes of *L. fabalis* and *L. obtusata* in different parameters. The metabarcoding method has already been successfully implemented for the molluscan bacteriome analysis. In particular, the impact of location and host species on freshwater mussel gut bacterial bacteriome composition [44]; the snails’ ability to accumulate new bacterial communities depending on the bacteria availability [45]; and the possibility of OTU-based metagenomic functional predictions on snails’ gut bacteriome [46] were convincingly demonstrated. In previous studies, we have shown the geographic and micro-niche variability in *Littorina* bacteriomes as well as the difference between these snails’ gut and body surface bacterial microbiomes and between associated and environmental ones [35,36]. In this project, we aimed at comparing the composition and spatial/temporal plasticity of the gut bacteriomes between the two sympatric sister-species in search of a possible role of commensal bacteria in host adaptation to their specific niches. We revealed significant between-season and between-region variability and described interspecies differences in the gut bacteriome composition. Particularly, we established several bacterial genera predominantly associated with one snail species and suggested hypothetical explanations for the adaptive significance of this enrichment.

## 2. Materials and Methods

### 2.1. 16S rDNA-Metabarcoding

#### 2.1.1. Sampling 

The study was designed to assess interspecies variation in the gut- and body-surface-associated bacteriomes of two sibling intertidal gastropod species, *Littorina obtusata* and *Littorina fabalis*, to identify seasonal and geographic variability in bacteriomes associated with different *Littorina* populations and their habitats. To assess the geographic variability of bacteriomes, sampling was performed in two regions: the Barents Sea, Dalnye Zelentsy and the White Sea, Kartesh cape. The samples from the Barents Sea region were collected at two sites: Oscar Bay (Barents#1, 69°07′01.0″ N 36°04′07.5″ E) and Yarnyshnaya Bay (Barents#2, 69°06′22.0″ N 36°03′35.0″ E) during the period 3–9 August 2021 (sampling period average water temperature (Ta) = 12 °C, average salinity (Sa) = 33.5‰; these data are consistent with typical values for the region [47]). In the White Sea region, sampling was performed at one site, Levaya Bay (66°20′16.7″ N 33°39′36.3″ E). To estimate the seasonal impact on bacterial microbiomes’ composition, the White Sea specimens were collected in different seasons: autumn (1–8 November 2021; sampling period average water temperature (Ta) = 2 °C, average salinity (Sa) = 24.5‰, these data are consistent with typical values for the region [48,49]) and early summer (1–7 June 2022; Ta = 10 °C, Sa = 24.8‰).

The samples were collected during low tide. Pooled environmental samples were obtained by numerous scrapings from natural substrates (fucoids and stones); a minimum of three replicates were made for each sample. The dissection of collected snails was carried out on the day of sampling in the field laboratory and included anatomical species identification [38] which was sufficient in this cryptic species group as a concordance between molecular data and anatomical features has been observed [50,51]. Each mollusk was dissected individually. This dissection procedure was practiced in the snail bacteriome studies [46,52]. For this process, two aseptically treated Petri dishes were used. In the first dish, soft tissues were taken out of the shell, and the shell fragments were washed away with sterile water. Then, in the second dish, genital morphology was first examined for species identification, then the tentacles were excised to be pooled from four individuals, and the resected intestinal fragments (midgut and hindgut) from every snail were fixed individually. Immature snails and individuals infected with trematodes were excluded from the analysis. After each dissection, Petri dishes and the instruments were routinely disinfected to avoid cross-sample contamination. To restrain any contamination, a negative control sample was obtained by using a sterile tampon to wipe the cleaned Petri dishes. All samples were fixed in 96% ethanol (environmental ones were fixed directly on the shore; snails’ samples were fixed during dissection).

#### 2.1.2. Library Preparation and Sequencing

Library preparation and sequencing were performed as described before [35,36]. In short: genomic DNA was extracted using PowerSoil Pro Kit (QIAGEN, Hilden, Germany) according to the manufacturer’s protocol. Then, a dual-indexing 16S rDNA library was prepared for sequencing on the Illumina MiSeq platform. Library preparation included locus-specific PCR, indexing PCR, and final pooling. Amplicons after either the inner or the outer PCR were cleaned-up with magnetic beads (Evrogen, Moscow, Russia). DNA concentration was measured using a Qubit 2.0 fluorometer (ThermoFisher Scientific, Waltham, MA, USA) with the QuDye dsDNA HS Assay Kit (Lumiprobe RUS, Moscow, Russia). The library preparation was based on the L. Hugerth protocol [53]. The 16S rRNA gene V4 region was amplified using 515F (5′–TGCCAGCMGCCGCGGTAA–3′) and 806R (5′–GGACTACHVGGGTWTCTAAT–3′) primers [54]. A total of 137 samples (Table 1 and Table 2) were sequenced on the Illumina MiSeq platform by Evrogen.

#### 2.1.3. Bioinformatic and Statistical Analysis

The reads were received demultiplexed, with adapter sequences trimmed. The quality of the data was evaluated using the FastQC version 0.12.0 tool [55]. The data were imported into the QIIME2 [56] denoising algorithm and amplicon sequence variants’ (ASVs) formations were conducted in DADA2 [57] wrapped as QIIME 2 plugin. Reads shorter than 250 bp, with number of expected errors higher than 2, and chimeras were discarded. The taxonomy of the representative sequences was determined using the sklearn-based classifier [58] trained on the sequences extracted from SILVA 16S rRNA gene database v. 138 [59]. Then, the data were filtered to exclude mitochondria and chloroplast sequences. Additional data filtering and normalization to the median library size was conducted in R (R Core Team 2021) using the phyloseq package [60].

To estimate alpha diversity, we used the Shannon–Wiener diversity and the Pielou evenness indices [61,62,63]. Beta diversity at the ASV level was visualized using the non-metric multidimensional scaling (nMDS) based on Bray–Curtis dissimilarity matrix [64]. The quality of ordinations was checked using the stress value [65]. Distances between the White Sea gut samples were additionally shown as boxplots grouped by host species and season. nMDS plots and boxplots were drawn in ggplot2 [66]. The effects of the host species, season, region and sample type factors on the bacteriome composition were tested, relying on permutational analysis using a linear model evaluation with a randomized residual permutation procedure (lm.rrppp function from the RRPP package (version 1.2.3) in R [67]) with the default parameters. The taxonomic composition of the samples was visualized, reflecting the names and relative abundances of the most abundant genera for samples averaged by sample type, on barplots built in the fantaxtic package [68]. We used the Random Forest classifier [69,70] in the MicrobiomeAnalyst web-service [71] to identify taxa important to particular sample compositions. For each pairwise comparison, a constant random seed (123456) was used and 500 decision trees were constructed to draw the final result. The relative abundance of the specific genera for each of the samples was visualized using barplots constructed in phyloseq. After statistical processing of the entire dataset, gut samples were subsampled and normalized separately to be run through the same analysis to avoid the loss of rare taxa and less prominent differences.

### 2.2. Shotgun-Sequencing and Metabolic Pathway Annotation

#### 2.2.1. Sampling

For each species, *L. fabalis* and *L. obtussata*, pooled fecal samples (feces of 15 individuals in one pool) were collected. Pooling material of several individuals is a generally accepted approach that allows for obtaining DNA concentrations sufficient for analysis [72,73,74]. Sampling was performed at the Oscar Bay site, the Barents Sea, in September 2022. Feces was collected via aseptic dissection, pooled in a tube for a species and fixed with 96% ethanol.

#### 2.2.2. Sequencing

DNA was extracted using PowerSoil Pro Kit (QIAGEN, Hong Kong, China) according to the manufacturer’s protocol. The DNA samples were transported to the Evrogen commercial service sequencing lab. The quality of genomic DNA was checked using agarose gel electrophoresis. Genomic DNA samples were prepared for sequencing using the Illumina DNA Prep kit. The quality of the resulting library pool was checked using an Agilent 2200 TapeStation instrument. Quantitative analysis of the pool was performed using qPCR. After quality control and DNA quantity assessment, the libraries were sequenced on the Illumina MiSeq device (read length 300 bp on both sides of the fragments) using MiSeq Reagent Kit v3 (600 cycles).

#### 2.2.3. Assembly and Annotation

The program FastQC (v. 0.12.0) [55] was used to check the quality of the obtained reads. Then, based on the quality results, the program fastp (v. 0.23.4) was used to remove sequences with total phred scores below 20, as well as adapter sequences [75]. Complete metagenomes were assembled using the SPAdes program in metaspades mode with a k-mer size of 23 and a minimum contig length of 200 [76]. Next, the program Kaiju (v. 1.9.2) was used for taxonomic classification [77]. The subset of NCBI BLAST nr database containing Archaea, bacteria and viruses (10 May 2023 data version) was used as the database. Taxonomic classification in Kaiju resulted in the generation of tables for the microbial genera and species associated with *L. obtusata* and *L. fabalis*. For each taxonomic unit in the table, information about its percentage composition relative to the entire sample, the number of reads mapped to the respective taxon, and its NCBI identifier were included. Species with an abundances of less than 0.01% were filtered out as noise. In the Kaiju output, for each target genus with a composition greater than 1% based on 16S data, corresponding species were identified. GCF (Genome Reference Consortium) files in fasta format were downloaded for each of these species. Subsequently, these files were annotated using the Prokka tool, v. 1.11 [78]. The program output yielded fasta files containing information on annotated protein sequences predicted from the genomic data. The annotation results for each genome were uploaded to the KEGG Automatic Annotation Server (KAAS) [79] for the prediction of metabolic pathways. Based on the predicted data, a table was compiled that provided information on the number of annotated genes related to the metabolic pathway present in the genome of the species of interest. At least two-fold differences in gene numbers involved in a particular metabolic pathway between genera of different comparison groups were accepted as significant (metabolic pathways represented by one or two genes were not considered).

## 3. Results and Discussion

### 3.1. General Patterns

The bacteriomes (bacterial microbiomes) from different sources were analyzed: the environmental biofilms (EB) from diverse living (algae) and non-living (stones) substrates of the intertidal zone, and the communities associated with body surfaces (BSB) and the guts (GB) of two periwinkle species, *L. fabalis* and *L. obtusata*. A comparison of the bacterial species’ compositions showed that the GB grouped mostly separately from the EB, while the BSB were located on the ordination in the middle zone between the GB and EB samples (Figure 1). Generally, this is consistent with the previously published thorough comparative analyses of the bacteriomes associated with the *Littorina* species [35,36]. In this study, we focused on the detailed comparison between the GBs of the cryptic sister-species *L. fabalis* and *L. obtusata*. Notably, the BSB and GB of both host species partially overlapped with the EB samples from *A. nodosum*, *F. serratus* and stones—the substrates where snails of both species can be found [42]. Only samples of *L. obtusata* tended to be ordinated closer to the EB of the *F. vesiculosus* from the middle shore level, where *L. fabalis* do not live (Figure 1). This corroborates the dependency of the composition of the snail-associated bacteriomes on the EB of a particular micro-biotope [35].

### 3.2. Sources of Variability in the Bacteriome Composition

We evaluated the variability of the GB composition in space and time: there were two collection sites in the same season (at the Barents Sea) and one site in two different seasons (at the White Sea) in our analysis. The dominant bacterial genera and overall community characteristics (species richness and evenness) remained consistent between the two Barents Sea sites. Moreover, there were no significant effects of the collection site (PERMANOVA *p* > 0.05). Notably, the differences between host species were found to be significant for bacteriome composition at the Barents Sea (PERMANOVA *p* = 0.012) (Figure 1 and Appendix A). The analyzed sites were not a long distance from each other and one of them (the Oscar Bay) was in close proximity to human habitation and to a docking area, which were expected to have impacts on coastal microbiota composition (e.g., high polyaromatic hydrocarbon concentrations were registered in the intertidal and subtidal sediment of the Oscar Bay, [80]). Accordingly, bacteria of the genus *Acinetobacter* were highly abundant in the EB and the BSB samples from the Oscar Bay exclusively; these bacteria were also registered in the GB samples of both host species, though were less abundant (Appendix A). The genus *Acinetobacter* is a common inhabitant of soil and ocean sediment, including polar regions, being enriched in the hydrocarbon-polluted sites [81]. Since the 1990s, it has also been recognized as an important infectious agent in humans worldwide [82], being associated with skin and gastrointestinal diseases and transmitted, e.g., via feces [83]. Thus, though some signs of anthropogenic impact in the Oscar Bay were registered, the scale of differences did not reach a significant level in this case.

On the White Sea coast, the gut bacteriome samples obtained from the same site in early summer and in autumn differed drastically (Figure 1 and Appendix A; PERMANOVA *p* = 0.001). These between-season compositional alterations were accompanied by a drop in the mean bacterial diversity (both alpha and beta diversity; Figure 2a,b); moreover, the bacteriomes’ compositional differences due to the host species became insignificant in autumn, though were detectable in early summer (PERMANOVA *p* < 0.01). Importantly, although the microbial diversity in the GB decreased in autumn, the dominant genera were detected in both seasons (Figure 2a and Figure 3). 

Studies on the seasonal variation in the gut bacteriome composition of marine invertebrates are still quite rare. Season was revealed to be a significant predictor of microbial community structure in guts of Eastern oysters *Crassostrea virginica* [84,85] and the urochordate ascidian *Halocynthia roretzi* [86], with the cold season being inferior to the summer months in the community richness in both cases. In the former case, the abundance of heterotrophic bacteria and carbon source utilization in the mollusk-associated bacteriomes correlated with the seawater temperature. The role of temperature as a factor structuring the bacteriome associated with the hemolymph of the *Crassostrea* oysters was also demonstrated in scales of temporal and spatial variation, as well as experimental conditions, with bacterial alpha diversity increasing along with warming [85,87,88]. Interestingly, we recalculated data published earlier [35], and revealed that the values of the gut bacteriome alpha diversity in three *Littorina species* (*L. littorea*, *L. saxatilis* and *L. fabalis*) tended to be lower in the northern collection sites (Norway, Tromso) compared to the southern ones (Sweden, Tjarno) (Figure 2c).

Obviously, temperature can affect the bacterial community, directly influencing its metabolic activity, production rate and assortment of organic compounds, and thus the bacterial diversity, which was demonstrated both within microbial mats and in plankton [89,90]. Such effects of environmental temperature are fairly expected also for bacterial communities associated with poikilothermic animals. Temperature also affects metabolic rates and moving activity (including foraging and grazing) of the poikilothermic hosts. The feeding activity of intertidal grazers has seasonality in temperate and polar latitudes. For instance, the *Patella* limpets demonstrated a clear decline in grazing activity during the cold season and its general positive correlation with the seawater temperature [91]. Seasonal patterns are also known in periwinkles; moreover, it was hypothesized that the winter decline in gastropod grazing activity may be a factor eliciting the recovery of algal abundances [41]. A decrease in feeding activity suggests a reduction in the engulfment of bacteria as a food substrate, which may impoverish the diversity of the gut-associated community. 

Another important point is that movement restraints caused by low temperatures affect the foraging activity of snails, limiting the diversity of microbiotopes where every particular individual can be encountered. Moreover, both *L. fabalis* and *L. obtusata* are prone to abandon the macroalgal canopy and hide at the base of the stones and rocks during the cold season [40]. That is, during autumn and winter, the two flat periwinkle species not only make contact with a lesser amount and a lesser diversity of EB bacteria, but also the sets of bacteria these species make contact with are more similar than during the summer season (due to the similarity of the occupied microbiotope). This explains why *L. fabalis* and *L. obtusata* harbor non-differing bacteriomes of relatively low diversity in the autumn samples. In addition, the cold-season-lowered foraging activity can explain the diminishing of not only the alpha but also the beta diversity observed in this study (Figure 2b). 

### 3.3. Associated Bacteriome Composition

The general composition of the gut bacteriomes associated with the *Littorina* snails was described in detail earlier [35,36]. Our present results on *L. obtusata* and *L. fabalis* corroborated the tendencies established previously for the *Littorina* (*Neritrema*) and *Littorina* (*Littorina*) species: The community is dominated by a limited number of bacterial lineages (3–5);The most abundant taxon is Proteobacteria, especially Gammaproteobacteria;Those less abundant, but inevitably present in the GB groups, are Fusobacteria (Fusobacteriales), Bacteroides (Flavobacteriales) and Planctomycetes (Pirellulales);Cytophagales are predominantly enriched in the BSB (Figure 3a).

The principal bacterial genera in the GB were *Psychromonas* (Gammaproteobacteria, Alteromonadales), *Psychrilyobacter* (Fusobacteria), *Vibrio* and *Aliivibrio* (Gammaproteobacteria, Vibrionales), and the less abundant *Pseudomonas* (Gammaproteobacteria, Pseudomonadales), *Blastopirellula* (Planctomycetes, Pirellulales), *Luteolibacter* (Verrucomicrobia), as well as the unclassified Alphaproteobacteria g. sp., Flavobacteria g. sp., Verrucomicrobia g. sp. and *Pirellulales* g. sp.

### 3.4. Interspecies Differences in the Gut Bacteriome Composition

During the summer season, we analyzed the between-species differences of the gut bacteriome in two geographic regions: the White Sea and the Barents Sea. We expected that in the White Sea the between-host-species differences in the associated bacteriome composition may be less pronounced compared to in the Barents Sea. We hypothesized this based on the differences between the Barents and the White seas in the distribution patterns of fucoids which the host snails are predominantly associated with. For instance, *F. serratus* (preferentially inhabited by *L. fabalis*) in the White Sea is present only sporadically and does not form zones of high density because it is sensitive to low salinity [92,93]. Indeed, we revealed distribution patterns for both *L. obtusata* and *L. fabalis* in the White Sea to be a little dissimilar to those in the Barents Sea. However, there still were clear ecological differences between them in both regions. In particular, in the White Sea, *L. obtusata* mainly inhabited the depth of the *F. vesiculosus* canopies (but *A. nodosum* in the Barents Sea, [42]), while *L. fabalis* was most often associated with *A. nodosum*, both on its surface and under the canopy (Appendix A), but preferred *F. serratus* in the Barents Sea [42].

Although, generally, the composition of high-rank taxa and the list of abundant genera in the GB of both host species were similar (Figure 3a,b), the statistically significant differences due to the host species were registered in both the White and the Barents Sea, which corresponds to the ecological differences described above. We applied the Random Forest analysis to identify some marker bacterial lineages strictly specific to the host species (exclusively detected in either *L. fabalis* or *L. obtusata* on a regular basis) and failed to find any. Yet, several bacterial genera tended to be enriched in one of the two host species, being the so-called soft markers (Appendix A). We considered only genera showing non-contradictory trends in different collection sites (even if not in all sites a particular genus was identified as a marker by the Random Forest). In this way, *Vibrio*, *Aliivibrio*, *Moritella* and *Planktotalea* were revealed as being predominantly associated with *L. fabalis*, while *Granulosicoccus*, *Octadecabacter*, *Colwellia*, *Pseudomonas*, *Pseudoalteromonas* and *Maribacter* were found to be primarily associated with *L. obtusata*.

We applied the Kaiju classifier to the shotgun DNA-sequencing data to establish the exact species in each target genus with relative abundances in the GB above 0.01 in at least one collection site. Then, based on the full bacterial lineage taxonomy, we performed an analysis of the enrichment of metabolic pathways in the soft marker genera compared to other abundant genera (≥0.01 mean relative abundance, with no clear distribution trends) using the number of genes involved in a certain pathway as a variable. Only one pathway (‘Phosphotransferase system PTS’, ID 02060), related to the uptake of carbohydrates, particularly hexoses, hexitols, and disaccharides, was found to be specifically enriched in bacterial genera associated with *L. fabalis*. In contrast, there were a number of pathways enriched in the bacteria predominantly present in the *L. obtusata* gut (Figure 4; Appendix A).

Among those pathways was, for instance, ‘Arachidonic acid metabolism’ (ID 00590). Interestingly, arachidonic acid was revealed through metabolomic profiling as being specifically abundant in the *L. obtusata* compared to other *Littorina* species [42]. The list of pathways revealed now as specifically enriched in the *L. obtusata* were ‘Fluorobenzoate degradation’ (ID 00364), ‘Benzoate degradation’ (ID 00362), ‘Aminobenzoate degradation’ (ID 00627), ‘Chlorocyclohexane and chlorobenzene degradation’ (ID 00361), ‘Styrene degradation’ (ID 643), ‘Toluene degradation’ (ID 00623) and ‘Xylene degradation’ (ID 00622). These pathways are associated with the genera *Pseudomonas* and *Granulosicoccus*; the former is most abundantly present in the *L. obtusata* GB of the White Sea samples, while the latter is in those of the Barents Sea; though both genera were registered in all populations analyzed (Appendix A).

The genus *Granulosicoccus* was reported from marine environments, including the seashore zone in the polar regions. It is known to be associated with brown macroalgae such as *F. vesiculosus* [94,95,96]; it was not previously reported in association with metazoans. *Granoulosicoccus* species are photoheterotrophs equipped with several metabolic functions of high relevance, such as nitrogen- and sulfur-transformation, the potential to synthesize cobalamin (B12), etc. [97]. Bacteria of the *Pseudomonas* genus are involved in antagonistic interactions and a variety of metabolic abilities related to synthesis of toxins and degradation of xenobiotics [98]. Species of the *Pseudomonas* genus were described in the gut-associated bacterial microbiomes of marine invertebrates [99,100]. The mentioned metabolic competencies rely on diverse pathways, including those listed above. Among the products/intermediates/substrates that can be processed in those metabolic pathways are phenolic and polyphenolic compounds [101,102,103]. This fact is of high importance with regard to the ecological differences between the host snail species *L. fabalis* and *L. obtusata*.

*L. fabalis* and *L. obtusata* demonstrate different, though partially overlapping, preferences to the shore levels, the host fucoid species and the placement in (on/under) an algal canopy [40,42]. One more feature differing in these two snail species is the structure of the radula, reflecting their divergence in feeding behavior [38,41]. While the diet of *L. fabalis* mainly includes epiphytic microbiota of the macroalgal surface, *L. obtusata* is able to excavate the depth of the seaweed thalli because its radular outer marginal teeth are armed with numerous angular cusps [40,41]. Brown algae (Phaeophyta) are well-known producers of diverse toxic phenolic and polyphenolic metabolites (e.g., phlorotannins), which defend them against fouling and grazing by micro- and macro-organisms [104,105,106]. Moreover, *F. vesiculosus* and *A. nodosum*, preferred by *L. obtusata*, contain higher concentrations of toxic polyphenols compared to *F. serratus*, which *L. fabalis* is predominantly associated with [41,107]. This poses the question about the mechanisms of tolerance of *L. obtusata* to the toxic action of tannins.

To our best knowledge, there are no data evidencing the ability of the *Granulosicoccus* or *Pseudomonas* bacteria (or their combination) to metabolize the phlorotannins of fucoids. Nevertheless, the well-established competencies of these micro-organisms to metabolically transform and degrade phenolic and polyphenolic compounds provides a basis on which to hypothesize their involvement in detoxification and conveying the adaptation of a host to its feeding practices. The possibility that the commensals of the *Pseudomonas* genus participate in the inactivation of the food toxins in the gut of the invertebrate herbivores was described in the example of the coffee berry borer beetle, *Hypothenemus hampei* [4].

The specific abundance of bacteria from genera *Granulosicoccus* and *Pseudomonas* in the intestinal bacterial community of *L. obtusata*, as well as the enrichment of polyphenol degradation metabolic pathways associated with these bacteria, may be a factor endowing these snails with the resistance to the toxic fucoids’ metabolites, such as phlorotannins, and allowing this species to occupy its specific niche. However, the details of the possible roles of *Granulosicoccus* or *Pseudomonas* in the degradation of food polyphenols in the guts of the *Littorina* snails are awaiting clarification in future studies.

## 4. Conclusions

We performed a detailed analysis of the bacteriomes associated with the cryptic sister-species of intertidal snails. Although there were significant inter-regional and inter-seasonal differences in the bacteriome composition, the genera dominant in abundance remained stable and similar between the host species. This is consistent with the idea of the *Littorina*-associated bacteriome conservatism stated previously.

We registered differences in the gut bacterial microbiome, body surface bacteriome and the environmental bacteriome, which were consistent with our previous results.

We revealed significant differences between *L. obtusata* and *L. fabalis* bacteriomes during the warmer season.

The decrease in the gut bacteriomes interspecific differences in autumn (and probably winter), as well as differences in alpha and beta diversity rates between seasons, indicates profound seasonal changes in the snails’ behavior. We expect these behavioral and compositional shifts to be less prominent in regions with warmer climates.

The predominant enrichment of the bacteriome with certain bacterial genera—especially for the *L. obtusata* gut bacteriome—is both a consequence of snail microhabitat features and their adaptations to the environment.

The enrichment of metabolic pathways involving the biodegradation of phenolic compounds in the genera *Granulosicoccus* and *Pseudomonas* predominantly associated with *L. obtusata* may be critical in mediating the tolerance of this species to fucoid-derived toxic substances (which *L. obtusata* adult snails engulf in a greater amount than their sister-species *L. fabalis*). In turn, such differences in bacteriome composition may act as a factor maintaining the differentiation of ecological niches between species.

## Figures and Tables

**Figure 1 biology-12-01521-f001:**
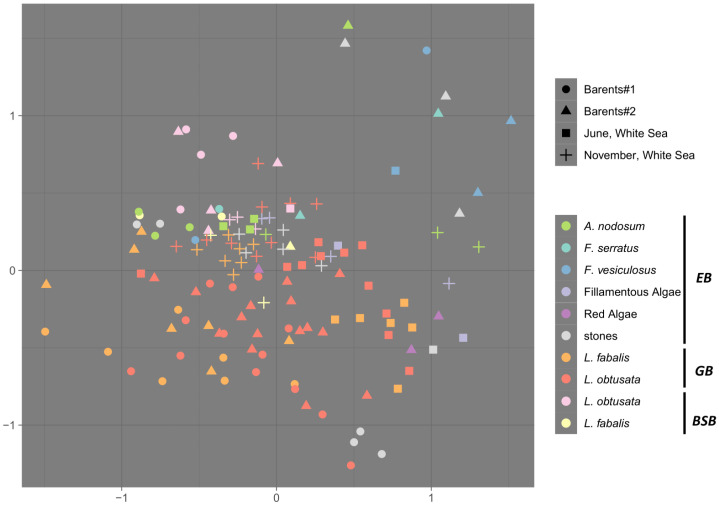
Comparison of bacteriome composition of all samples. nMDS ordination of the bacteriome in different sample types. EB—environmental bacteriome, GB—gut bacteriome, BSB—body surface bacteriome.

**Figure 2 biology-12-01521-f002:**
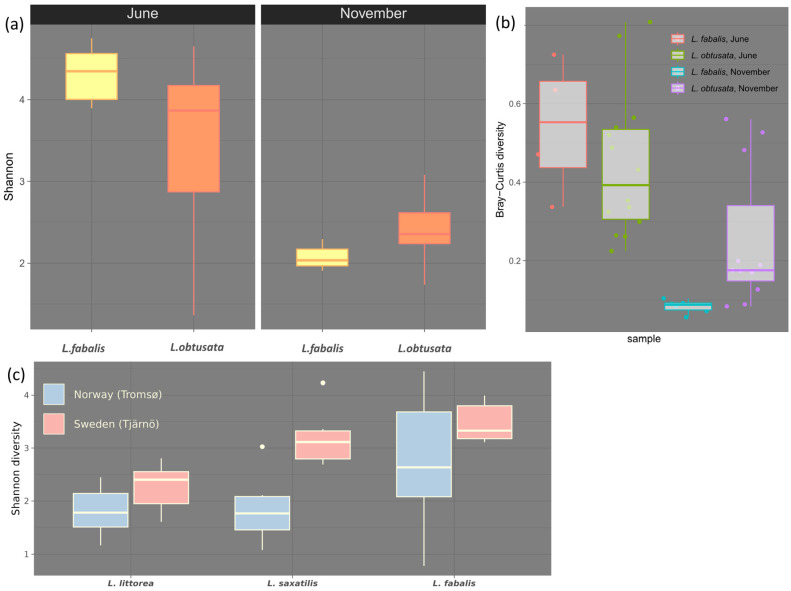
Diversity overview. (**a**) Shannon diversity index values. The index was calculated for the White Sea samples on the subsampled gut dataset. Index values for samples from a host species are combined into a boxplot. (**b**) Beta diversity measures. Bray–Curtis dissimilarity indices for the White sea samples. Distances were calculated on the subsampled gut dataset and shown as boxplots grouped by host species and season. (**c**) Shannon diversity index values (previously published data [32]). The general trend of the associated bacterial community alpha diversity value to decrease and can be demonstrated by latitudinal comparison. The Shannon diversity index values for our previously published data on three *Littorina* species (*L. littorea*, *L. saxatilis* and *L. obtusata*) are in consistency with this trend.

**Figure 3 biology-12-01521-f003:**
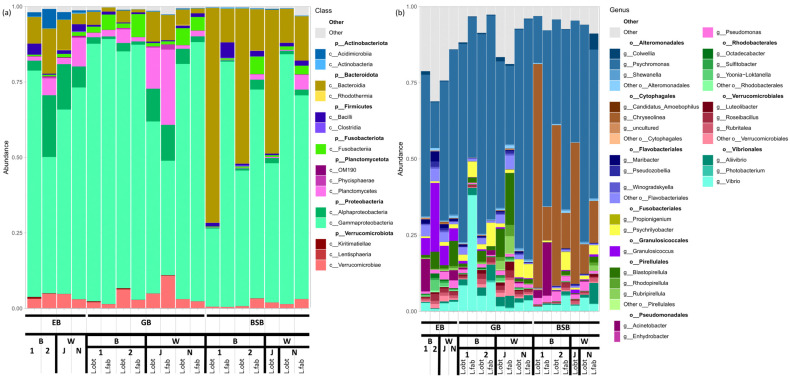
Relative abundance of the most abundant bacterial taxa averaged for each sample type. (**a**) Relative abundance at the class level. Bacterial classes are grouped into phyla. (**b**) Relative abundance at the genus level. Bacterial genera are grouped into orders.

**Figure 4 biology-12-01521-f004:**
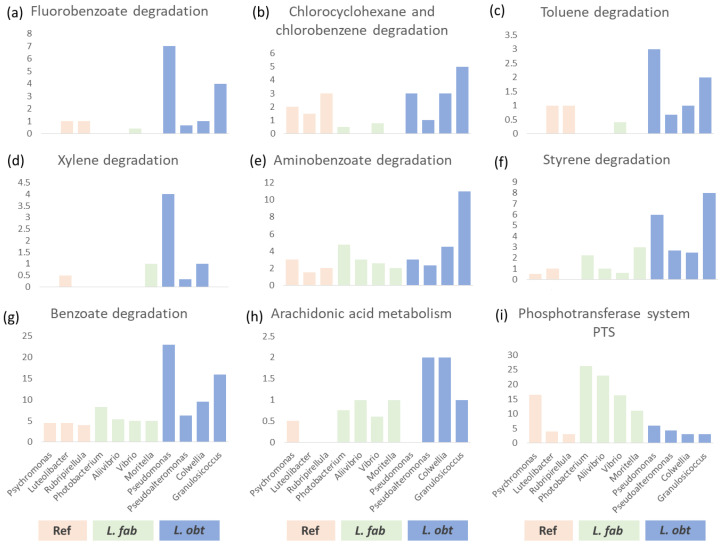
The mean number of genes involved in particular metabolic pathways. (**a**) Fluorobenzoate degradation. (**b**) Chlorocyclohexane and chlorobenzene degradation. (**c**) Toluene degradation. (**d**) Xylene degradation. (**e**) Aminobenzoate degradation. (**f**) Styrene degradation. (**g**) Benzoate degradation. (**h**) Arachidonic acid metabolism. Interestingly, a slight trend for genes of this pathway to be enriched in *L. obtusata* is compatible with the results of a recent metabolomics study, where arachidonic acid was identified as a compound specifically enriched in *L. obtusata* [42]. (**i**) Phosphotransferase system (PTS). The numbers of considered species were in the reference group (ref)—*Psychromonas*: 2 (*P. ingrahamii*, *P.* sp. *CNPT3*); *Luteolibacter*: 2 (*L. ambystomatis*, *L. luteus*); and *Rubripirellula* 1 (*R. lacrimiformis*). In the group of species predominantly present in *L. fabalis* (*L. fab*)—*Photobacterium* 4 (*P. gaetbulicola Gung47*, *P. damselae*, *P. profundum*, *P. ganghwense*); *Aliivibrio* 3 (*A. fischeri ATCC 7744*, *A. fischeri MJ11*, *A. salmonicida*); *Vibrio* 5 (*V. breoganii*, *V.* sp *VB16*, *V. algicola*, *V. ciclytrophicus*, *V. splendidus*); and *Moritella* 1 (*M. marina*). In the group of species predominantly present in *L. obtusata (L.obt)*—*Pseudomonas* 1 (*P. stutzeri*); *Pseudoalteromonas* 3 (*P. carrageenovora*, *P.* sp. *A25*, *P. tunicata*); *Colwellia* 1 (*C. psychrerythraea*); and *Granulosicoccus* 1 (*G. antarcticus*).

**Table 1 biology-12-01521-t001:** A list of the successfully sequenced samples collected on the Barents Sea coast.

Sample	Barents#1	Barents#2
Environment	*F. vesiculosus* scraping (×2)	*F. vesiculosus* scraping (×2)
	*A. nodosum* scraping (×3)	*A. nodosum* scraping (×1)
	*F. serratus* scraping (×1)	*F. serratus* scraping (×2)
	Stones scraping (×5)	Stones scraping (×3)
		Red Algae scraping (×3)
Snails	*L. fabalis* (gut ×7, tent. ×2)	*L. fabalis* (gut ×8, tent. ×2)
	*L. obtusata* (gut ×13, tent. ×4)	*L. obtusata* (gut ×15, tent. ×4)

**Table 2 biology-12-01521-t002:** A list of the successfully sequenced samples collected on the White Sea coast.

Sample	White, June	White, November
Environment	*F. vesiculosus* scraping (×1)	*A. nodosum* scraping (×3)
	*A. nodosum* scraping (×3)	Filamentous Algae scraping (×4)
	Filamentous Algae scraping (×2)	Stones scraping (×5)
	Stones scraping (×1)	
Snails	*L. fabalis* (gut ×6)	*L. fabalis* (gut ×7, tent. ×2)
	*L. obtusata* (gut ×11, tent. ×1)	*L. obtusata* (gut ×12, tent. ×3)

## Data Availability

Raw data are available for downloading from the NCBI database. 16S rRNA gene metabarcoding reads are deposited as PRJNA1044559 and shotgun metagenomics raw sequences are deposited as PRJNA1044926.

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
