# Peer review of "Gut Bacteriomes and Ecological Niche Divergence: An Example of Two Cryptic Gastropod Species"

_biology, 2023, doi:10.3390/biology12121521_

Round 1
Reviewer 1 Report
Comments and Suggestions for Authors
Review comments
The MS explores the differences between two types of snail gut microbiota in different seasons and environments based on 16S rRNA and metagenomic sequencing technology, but the research results are not clearly presented, making the content difficult to understand.
The expression of professional terms in microbial sequencing is not accurate enough. There were also many details to consider.
Abstract:
The key content in the abstract is not prominent. Does the article only identify seasonal differences and hypotheses based on correlation analysis? And there is also no explanation of the results related to the research on ecological niche divergence in the title. The conclusion also has the same problem.
Introduction:
Lines 11,20,39,44…: “gut or intestinal”, please unify the expression of words.
Lines 52-53: Insufficient theoretical basis for the important role of gut microbiota in host adaptation to ecological niches
Line 74: “16s-metabarcoding” ?
Materials and methods:
The six biological replicates of environmental samples in Line 91 are inconsistent with the representations in Tables 1 and 2, and how to perform correlation analysis between environmental samples and snail samples.
Line 150:If a mixed sample of 15 individuals is used for the metagenomic sample, the reliability of the sequencing results will be greatly reduced.
Results:
The processing group of the MS includes different environmental samples, snail body surface, and gut microbiota. What is the relationship between the differences in microbial communities among these treatments and niche adaptation? How to analyze this ecological adaptation mechanism using metagenomic sequencing instead of drawing hypothetical conclusions.
Line 201: Incorrect citation format of reference
Reviewer 2 Report
Comments and Suggestions for Authors
The research work of Gafarova et al. is in line with the journal's focus areas.
Although Results part authors use the correct term "bacteriomes", "microbiome" is very often mentioned in the title, summary and abstract. Correction required.
The introduction lacks references to known authors using similar methods in studies of other molluscs/snails (DOI: 10.1371/journal.pone.0224796, 10.1093/femsec/fiad101 or 10.7717/peerj.5537).
Why did the methods not use molecular identification of the molluscs studied, only anatomical identification?
Why don't the authors hereafter use the modern taxonomy of microorganisms presented for example here - https://gtdb.ecogenomic.org/?
Required to improve the quality of Figures 2, 3 and 4.
The manuscript requires the addition of an explanation of a phrase from Abstract “We hypothesize that the L. obtusata gut microbiome contributes to decomposing of algae and detoxifying of polyphenols produced by fucoids“.
Reviewer 3 Report
Comments and Suggestions for Authors
The manuscript "Gut microbiomes and ecological niche divergence: an example of two cryptic gastropod species" by Elizaveta Gafarova, Dmitrii Kuracji, Karina Sogomonyan, Ivan Gorokhov, Dmitrii Polev, Ekaterina Zubova, Elena Golikova, Andrey Granovitch and Arina Maltseva is devoted to the study of gut microbiomes of closely related marine snail species taking into account the influence of spatial and temporal factors.
After careful reading and judgement, I recommend that the manuscript be thoroughly revised and then resubmitted. The comments below are intended to help the authors revise their manuscript:
1. The Introduction does not have a clearly stated purpose.
2. Also, more information about the snail species used in this paper should be added to the Introduction of the manuscript. Why do the authors believe that these two species can be used as a model to understand the drivers, mechanisms and consequences of ecological speciation?
3. The authors should explain why they sampled at different times in different seas. Why didn't they just sample in August? Or why the effect of seasonality on the gut microbiomes of L. fabalis and L. obtusata was not also studied in the Barents Sea?
4. In the Materials and Methods section, the water temperature at the time of sampling should be reported, as well as the climatic characteristics prior to sampling (in this case, authors should provide a link to an Internet meteorological resource with a weather archive). Information on the salinity of the two seas, which is also different, should also be included. Salinity affects microbial diversity, which in turn may affect the gut microbiome of L. fabalis and L. obtusata. These data are even more necessary as the authors also mention the role of salinity in the distribution of F. serratus.
5. Were the snail shells treated before dissection? How was contamination of gut samples with microorganisms from the shell surface avoided? This information should be added to the manuscript.
6. Sampling in the White Sea was carried out in the summers of 2021 and 2022. The remaining samples were only collected in 2021. In this case, it is completely incorrect to combine the microbiome analysis data of summer samples from different years. The authors should present the microbiome analysis data of the samples collected in the summer of 2022 separately and compare the results with the data of the samples collected in the summer of 2021 (according to the weather archives in 2021 and 2022, the meteorological conditions in the period of sampling and in the period before sampling were different. For example, ambient temperatures in May and early June were sometimes higher in 2021 than in 2022. In addition, the influence of the "winter" factor should not be excluded. Winter conditions in 2020-2021 and 2021-2022 may differ not only in the temperature component but also in other factors affecting water surface conditions at the sampling site). Data from summer 2022 samples should be excluded from the bioinformatics results presented in the manuscript.
Minor comments
7. Line 216 - Figure 2 refers to microbiome analysis of samples from the White Sea, but this sentence only mentions 2 sampling sites in the Barents Sea, corresponding to Figures 1a and 1b. The captions of Figures S1 and S2 should also clarify that the letters e, g, tent stand for "environment", "gut" and "tentacle" respectively (Did the reviewer understand this correctly?). Authors should check that the order of figure references in the supplementary material is correct.
8. Lines 236-238 should be deleted.
9. Lines 368-370 do not refer to figure captions.
Comments on the Quality of English LanguageMinor editing of English language required
Round 2
Reviewer 1 Report
Comments and Suggestions for Authors
In the 16S rRNA sequencing analysis, the author selected microbial research factors such as season, environment, body surface, and gut, but only extracted the influence of seasonal factors in the abstract and summary. What are the conclusions of other factors? What are the differences between different species (not just a few species)? Metagenomics can use mixed sampling methods, but it is necessary to ensure 5 or more biological replicates! Is the metagenomic analysis result appropriate under the title of 3.4? Line 105:16s should be 16S.
Author Response
Thank you for your review. We appreciate the time and effort you have taken to consider our work. Please find our responses below and the corresponding corrections in the MS.
Comment 1. In the 16S rRNA sequencing analysis, the author selected microbial research factors such as season, environment, body surface, and gut, but only extracted the influence of seasonal factors in the abstract and summary. What are the conclusions of other factors? Metagenomics can use mixed sampling methods, but it is necessary to ensure 5 or more biological replicates! Is the metagenomic analysis result appropriate under the title of 3.4.
Response 1. Thank you for your comment. As you rightly point out, the number of replicates for environmental and body surface samples does not allow us to draw confident conclusions regarding the difference of these samples and gut microbiomes. This is the reason we focus on seasonal, geographic and interspecies differences in bacterial microbiomes in the abstract and conclusions. Chapter 3.4 in the results represents the analysis of the gut microbiomes of two gastropod species as it is specified in its title (Interspecies differences in the gut bacteriome composition). For the gut samples, both the replicates number and the analysis methods allow the discussions and conclusions outlined.
We have added the clarification to the first sentence of the section 3.4 (lines 352-353): “During the summer season, we analysed the between-species differences of the gut bacteriome in two geographic regions: the White Sea and the Barents Sea”.
Nevertheless, we consider a qualitative description of the environmental bacteriome and the bacterial microbiome of the body surface of molluscs to be acceptable and leave it reflected in section 3.1 General patterns. Besides, these results resemble the ones obtained in previous studies (doi:10.1371/journal.pone.0260792, doi:10.1111/eva.13447).
We have extended the reference to our previous results in the Introduction section (lines 97-99): “In previous studies, we have shown the geographic and microniche variability in Littorina bacteriomes as well as the difference between these snails’ gut and body surface bacterial microbiomes and between associated and environmental ones”.
We have added the additional conclusion paragraph (lines 465-466):
“We registered differences in the gut bacterial microbiome, body surface bacteriome and the environmental bacteriome, which is consistent with our previous results”.
Comment 2. What are the differences between different species (not just a few species)?
Response 2. The study is aimed at estimating the differences between the two cryptic species of intertidal molluscs. The title of the work and the abstract (lines 24-26) include this claim. We have added clarifications to the summary (line 13) and methods (line 111) that we are comparing 2 species of gastropods: Littorina obtusata and L. fabalis.
Comment 3. Line 105:16s should be 16S.
Corrected.
Reviewer 3 Report
Comments and Suggestions for Authors
All reviewer's comments and recommendations have been taken into account by the authors of the manuscript.
Author Response
Thank you for the review. We appreciate the time you have taken to consider our work. We express our sincere gratitude to you for your suggestions, which have contributed to enhancing the quality of our manuscript significantly.